# Temporal regularities of vocal exchange in Java sparrows

**Sota Kikuchi** [ORCID], **Noriko Kondo**, **Hiroki Koda** [ORCID] *

1 Graduate School of Arts and Sciences, The University of Tokyo, Komaba, Tokyo, Japan

☉ These authors Co-first authors/equal contributions
* hkoda@g.ecc.u-tokyo.ac.jp

## Abstract

In human turn-taking, speakers alternate at very short intervals while avoiding overlaps. If speakers do not receive a vocal response from another party, they often repeat their utterance after the expected response time has elapsed to elicit a reply. Intra-individual intervals tend to be longer than inter-individual intervals. Such temporal regularity in vocal exchanges has also been observed in social mammals, such as non-human primates. In contrast, vocal exchanges in birds have been studied primarily in the context of songs, with limited research on call-based vocal exchanges. Studies specifically examining intra-individual call intervals are even scarcer. In this study, we investigated vocal exchanges and their temporal patterns in Java sparrows placed face-to-face. The results revealed that they vocalized at very short intervals following the calls of the other individual. However, no significant differences were observed between the inter- and intra-individual intervals. This contrasts with the temporal characteristics of vocal exchanges observed in social mammals that have been studied to date. We propose that a possible explanation for this difference lies in the variation in social group structures between birds and mammals.

## Introduction

Human conversation is characterized by a vocal turn-taking phenomenon, in which speakers quickly alternate roles to convey information to each other (e.g., [1]).

Turn-taking in conversations is characterized by two key features: i) the preservation of extremely short intervals between turns, and ii) the avoidance of overlapping speech. Notably, inter-individual intervals–the time between the end of one speaker's utterance and the beginning of the next–exhibit a universal temporal regularity, averaging approximately 200 ms regardless of linguistic or cultural differences [2,3].

As human vocal turn-taking is characterized by consistent short temporal regularity regardless of semantic content, the vocal exchange of non-human animals can be examined within the same framework. Numerous studies have explored vocal interactions across various animal species in an effort to draw parallels

**Data availability statement:** The data on which this study is based are available from https://doi.org/10.6084/m9.figshare.29329889

**Funding:** We were supported from: Japan Society for the Promotion of Science KAKENHI (B, #23K25168/22H03914, https://kaken.nii.ac.jp/en/grant/KAKENHI-PROJECT-23K25168; S, #23H05428, https://kaken.nii.ac.jp/en/grant/KAKENHI-PROJECT-23H05428) awarded to HK, Collaborative Research Program of Wildlife Research Center, Kyoto University (2024-A-016) awarded to NK (https://www.wrc.kyoto-u.ac.jp/en/joint-research/index.html) The funders listed below had no role in study design, data collection and analysis, decision to publish, or preparation of the manuscript.

**Competing interests:** The authors have declared that no competing interests exist.

with human conversational turn-taking [4,5]. Primate vocal exchanges have been extensively studied, particularly in *contact calls*. *Contact calls* are common vocalizations exchanged between individuals in a cohesive social group and have been analyzed in detail to characterize the intervals in vocal exchange. Similar to human conversation, vocal exchanges in many primate species exhibit temporal regularity, characterized by stable intervals between individuals' vocalizations and significant avoidance of overlap (e.g., [6–9]). In addition, it has been reported that the time interval for repeated vocalizations by the same individual in the absence of a response from other individuals (repeated transition) is longer than the time interval for vocalizations by different individuals taking turns (exchange transition) [8,10–15]. A longer time interval in the exchange transition can be interpreted as the manifestation of a vocalizing individual expecting a response from another individual. In other words, after calling, the caller anticipates and waits for a response from another individual, and if no response is received, it calls again to elicit a reaction. This temporal regularity represents a key similarity between human conversation [16–18] and animal vocal exchange and has been reported in socially living non-primate mammals, such as rodents [19,20], bats [21], elephants [22,23], meerkats (*Suricata suricattal*) [24], and cetaceans [25–30]. Accordingly, this can be considered a common rule in the communication of social mammals and is adaptive in socially structured groups. (See the conceptual framework of temporal regularity in Fig 1.)

When social mammals are engaged in a reciprocal vocal exchange (A, top), the individual expects a "response" from the other individual and therefore waits for a response. After the expected response time had elapsed, the individual repeated the

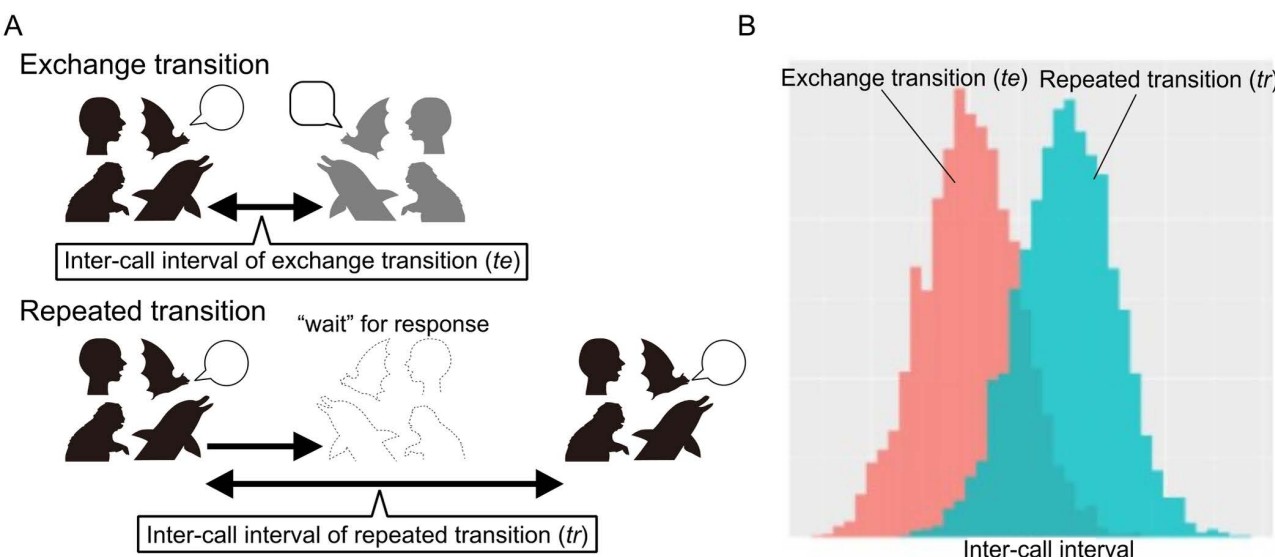

**Fig 1. Conceptual representation of the mechanism underlying the emergence of temporal regularity in vocal exchange/turn-taking of various social mammals.**

vocalization (A, bottom). Fig 1B illustrates the inter-call intervals of vocal exchange in mammals. The inter-call interval of the repeated transition $t_r$ is longer than that of the exchange transition $t_e$ (A, B).

Studies of vocal communication in birds have mainly focused on songs, the elaborate and complex vocalizations that function in territorial defense and mate attraction [31]. Song interaction studies include male-male counter-singing [32] and male-female duets [33]. The exchange of calls, on the other hand, has not been as extensively documented as in primates. However, calls are more appropriate than songs and duets when compared with human conversational vocal turn-taking and mammalian vocal exchange, as the latter are primarily restricted to breeding contexts. Some studies have examined the vocal exchange in bird species from a comparative perspective with that of mammals. In large-billed crows (*Corvus macrorhynchos*), temporal regularity in the vocal exchange of their contact, *'ka call'* - a fixed time exchange interval between different callers – was observed [34]. Similarly, the vocal exchange of calls in parents and offspring in common starlings (*Sturnus vulgaris*) [35] and the begging calls of barn owls (*Tyto alba*) [36] also confirmed the avoidance of caller vocal overlap, indicating temporal regularity in these call exchanges, although these studies did not examine the inter-call intervals in detail. Recently, calls have received more attention, and call exchange has begun to be studied in zebra finches (*Taeniopygia guttata*) of the family Estrildidae, a leading model species in song research, providing new insights into avian vocal exchange. Experimental studies examining vocal call exchange in zebra finches placed face-to-face confirmed that call overlap was significantly avoided and vocal timing was actively adjusted [37–40]. The authors of these studies argued that the vocal timing adjustments observed in zebra finches, which depend on the timing of other individual calls, are analogous to primate vocal exchanges. Such adjustments in call timing are considered a reflection of social relationships, such as pair mating, in zebra finches [40], suggesting that ecological adaptations in vocal interactions are similar to those observed in mammals. Although these findings highlight the role of vocal timing adjustments in social interactions, research to date has primarily focused on the intervals between calls produced by different individuals, while studies on the intervals between repeated transitions remain limited (see [41]).

When considering the role of vocal turn-taking in animal societies, the socio-ecological structure of the species in question should be taken into account. Social mammals form stable groups with fixed membership even outside the breeding season, often including unrelated individuals. In such groups, vocal turn-taking helps maintain group cohesion by enabling individuals to signal their location and avoid separation through reciprocal vocal responses [42]. Birds, on the other hand, also form social flocks, but these tend to be more fluid and less stable than mammalian social structures. Within bird flocks, the most salient social units are often pair-mates, family members, or kin. For example, species such as Greylag geese (*Anser anser*) form matrilineal alliances [43], yet they do not necessarily maintain exclusive territories with fixed group membership, as is often observed in mammal groups. Given these socio-ecological differences, along with the fact that birds in general are capable of traveling much faster than mammals due to their ability to fly, temporary spatial separation from flock members likely poses a lower risk for birds than for group-forming social mammals. Accordingly, vocal turn-taking with an expectation of a response appears to be less essential for birds than for social mammals as a mechanism for maintaining proximity and group cohesion. As is evident from recent zebra finch studies, however, the vocal exchange of calls in birds has been increasingly examined using a framework similar to that applied to primates. Previous studies have shown that humans, non-human primates, mammals, and birds share common vocal exchange features—specifically, short inter-individual intervals and longer intra-individual intervals, both of which help to avoid call overlap. Due to these shared characteristics, it has been hypothesized that vocal exchange may serve comparable functions across taxa. Nevertheless, as noted above, fundamental socioecological differences between social mammals and birds may shape their communicative strategies in distinct ways. Importantly, most avian vocal exchange research has focused on zebra finches, a key model species in songbird studies. These studies, which involved controlled face-to-face experiments, demonstrated significant avoidance of vocal overlap and active adjustments in call timing [37–40]. The authors of these studies proposed that such timing adjustments, which depend on the calls of others, are analogous to primate-like vocal turn-taking. Moreover, call timing was found to reflect social relationships such as pair bonding [40], suggesting that ecological functions of call timing

may be more similar across taxa than previously assumed. However, these studies have mainly focused on inter-individual call intervals, and research on intra-individual repeated transitions remains limited (see [41]).

In this study, we investigated the temporal regularity of vocal exchanges in the Java sparrow, *Padda oryzivora*. Java sparrows are small birds belonging to the family Estrildidae that are native to Java and Bali, Indonesia. These are socially monogamous species that sometimes form large flocks [44]. Their calls have not been well studied, except for trill calls, which are given in the context of affiliative and agonistic interactions [45]. There are also short monosyllabic calls, presumed to function as contact calls [44], and we focus on the exchange of these calls. As Java sparrows belong to the Estrildidae family, as do zebra finches, we expected to observe vocal exchange with characteristics similar to those of zebra finches, that is, short intervals between different birds. In addition, we investigated the within-individual call intervals to test whether Java sparrows avoid the temporal window in which a 'response call' from the other individual is most likely to occur, resulting in longer call intervals of the same individual. More precisely, we investigated the temporal regularity of their vocal exchanges by experimentally arranging multiple individuals face-to-face in a soundproof chamber, similar to the experimental settings of studies on marmosets and zebra finches [8,11,46]. We compared the time intervals between calls exchanged between different individuals and those repeated by the same individual to examine the characteristics of their temporal distribution.

## Methods

### Subjects

Six adult Java sparrows (five males and one female, 2–6 years of age) were used in this study. All individuals were housed separately in identical cages (38 cm high × 31 cm wide × 25 cm deep) within the same housing area, where they could hear and see each other. The housing environment was maintained with a 12-hour light:12-hour dark (12L:12D) lighting cycle, a room temperature ranging from 20°C to 28°C, and humidity levels kept above 40%. Adequate amounts of food and water were provided in each cage, and the subjects had unrestricted access to them throughout the experiment.

### Apparatus

The experiment was performed in a soundproof chamber (2146 cm H × 1378 cm W × 996 cm D) placed in a room separate from the housing room. Within the chamber, we designed an experimental space that allowed individual cages to be placed opposite to each other, facing one another (Fig 2A). We placed the two cages at two different locations where the two target birds could audio-visually recognize each other. The horizontal distance between the two cages was 58 cm. We attached two small microphones (CVM-V02C, COMICA), one directly above each of the two cages (approximately 10 cm), to allow close-up recording of vocalizations in each cage. The microphones were connected via cables to an audio interface (UMC1820 U-PHORIA, Behringer) using a computer (Ubuntu 20.2, CPU: Intel Core 5, RAM 16GB) located outside the soundproof chamber. The audio signals were recorded at a 32-bit quantization/32 kHz sampling rate.

(A) The experimental flow of the three phases, along with cage dimensions and spatial arrangement. (B) The sound spectrogram (top), waveform (middle), and segmental call regions labelled by evaluators (bottom), obtained from a 2-channel recording. (C) Workflow to generate pseudo turn-taking data (right). The originally recorded F2F data were replaced in different recording sessions, keeping the individual pair relationship consistency.

### Procedures

A single session of the experiment consisted of three phases (pre-face-to-face phase, PRE phase; face-to-face phase, F2F phase; and post-face-to-face phase, POST phase) according to the following procedure (Fig 2A).

First, the animals were exposed to a 15-minute PRE phase; in the PRE phase, only one individual (the subject) was introduced into the soundproof room and placed in an individually compartmented area. The purpose of the PRE phase was to

habituate the participants to the experimental space and encourage spontaneous vocalizations throughout the experimental session while reducing their stress. The duration of the F2F phase was 15 minutes. In the F2F phase, the other individual (facing individual) was placed opposite the subject bird facing it. Finally, the POST phase was set for 15 min. During the POST phase, facing individuals were removed from the soundproof room with only the preceding individuals remaining, as was the case during the PRE phase. Four sessions were performed for every combination of two individuals selected from six individuals (15 different pair sets) for a total of 60 sessions (2700 min) of recording. The order of the pair sets was randomized as much as possible to make sure that the same individuals did not participate in consecutive experimental sessions.

### Acoustic analysis

**Labelling vocal data.** Before analysis, all vocalizations recorded during the experimental sessions were labelled (or annotated). Labelling was performed using three processes: segmentation, classification, and caller identification. All labelling tasks were shared by two independent evaluators (SK and NK) using Raven Pro version 1.6 [47] or Praat version 6.3.10 [48]. Before the labelling tasks, the two evaluators checked with each other to agree on the criteria for the segmentation and classification of vocalization types (Fig 2B).

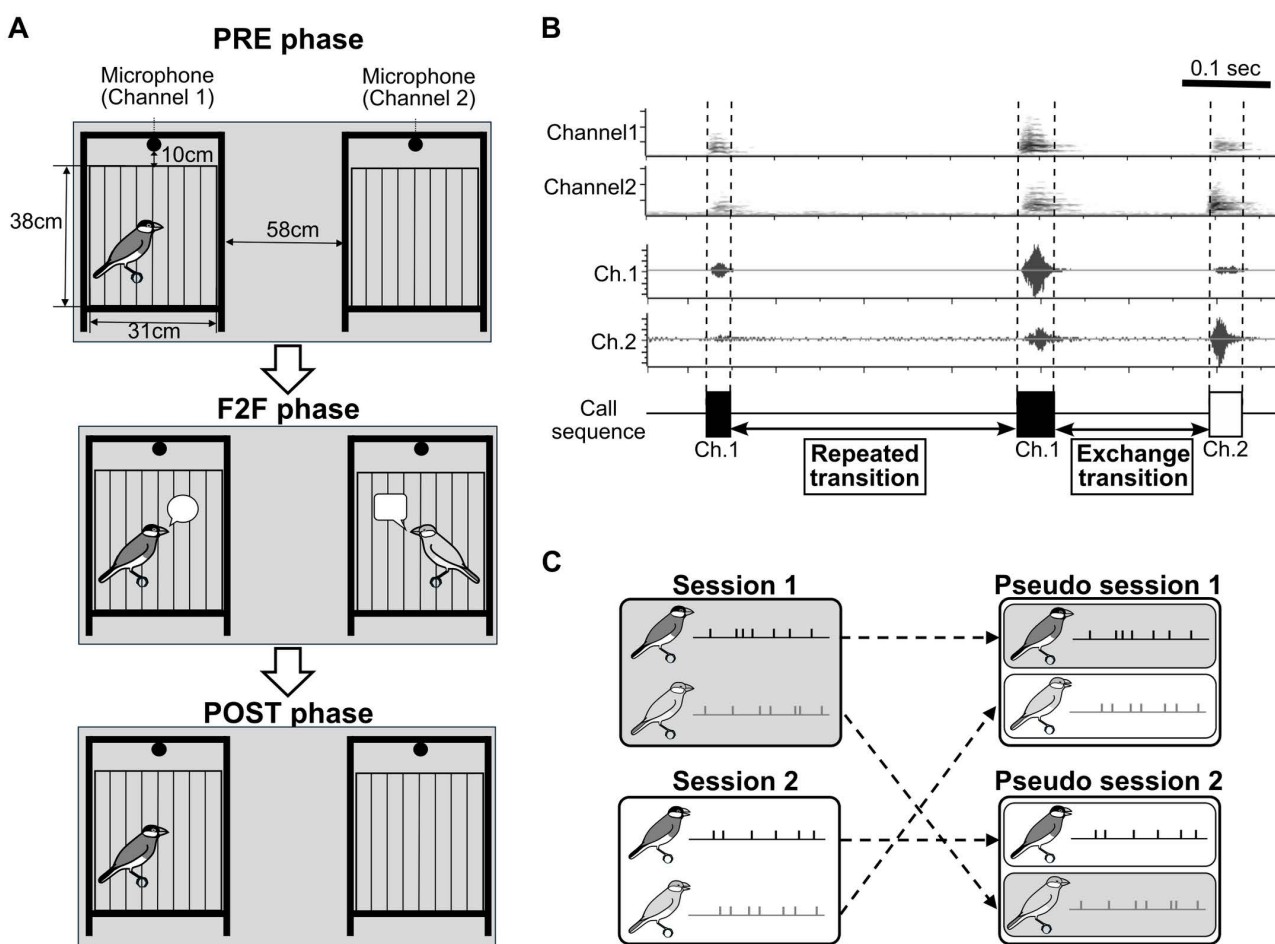

**Fig 2. Schematic illustrations of (A) experimental settings, (B) labelling procedure of calls, and (C) the generation process of shuffled session data.**

## Segmentation

Two evaluators independently judged the intervals between vocalizations to define the intervals in the vocal region. Specifically, the onset and offset times of the vocal regions were logged from the recorded data based on the visual inspection of the waveforms and spectrograms while listening to the audio data.

## Classifications and caller identifications

The detected vocal regions are classified into three types of vocal types (calls, songs, and trills). Here, *songs* were classified as "vocal sequences that appear in a stereotyped pattern for an individual," *calls* as "monosyllabic and short-note vocalizations that have no sequential rules and are not songs," which is described as "contact calls" in [44], and *trills* as "typical sounds that appear as short-duration repetitive pulses."

Caller identification was performed by comparing the intensities of multiple channels in the recorded data. The location of the microphone channel with higher intensity was judged to be closer to that of the target individual.

## Labelling validation

To check the validity of the labelling, a subset of the data was extracted: four sessions of data were selected from 60 sessions to include approximately 1000 vocalizations, and both evaluators independently labelled the data in the F2F phase. The "agreement" of the labelling between the two labelled data (label-SK, label-NK) was examined. The agreement rate between two annotation labels (label-SK and label-NK) for one segment was defined as the accuracy rate and was calculated as follows: $Acc = \frac{N_{agree}}{N_{all}}$, where $Acc$ represents the accuracy rate, $N_{all}$ is the total number of segmented vocal regions, and $N_{agree}$ is the number of agreements.. The accuracy rate for classification and caller identification was 95.75% (of 969/1012 vocalizations). We also examined whether the onset and offset times differed between the evaluators. The onset and offset times were recorded in the two labelled datasets (label-SK and label-NK). To compare the difference in time, we calculated the time difference ($\Delta t := t_{sk} - t_{nk}$) from the time in label-SK ($t_{sk}$) to the time in the nearest neighbor label-NK ($t_{nk}$), for onset and offset, respectively. The interval estimation of the obtained time difference data (n = 969) was performed using a simple one-sample t-test. If there was a decision bias to annotate label-SK earlier (or later) than label-NK, then there would be a "biased" interval estimation that did not include $0$. Otherwise, the interval estimation includes $0$. The t-test results estimated the intervals to be [−0.006, 0.010] and [−0.025, 0.012] for the onset and offset times, respectively, with no statistically significant value bias compared to 0. Accordingly, this result indicates that there was no statistical judgment bias for label-SK and NK.

## Definition of the call transition sequences

For the analysis of vocal exchange, we only used recordings of the F2F phases. We define the type of call transition sequence as follows: An "exchange transition" was defined as a case in which a vocalization by one individual is followed by a vocalization by a different individual, and a "repeated transition" as two consecutive vocalizations by the same individual. In exchange transitions, the offset time of the preceding vocalization may be later than the onset time of the following vocalization when an overlap occurs. Following these definitions, we defined a "call interval" as the time interval between the offset of the preceding call and the onset of the subsequent call, regardless of the vocalizing individual.

## Statistical procedures

**Estimations of interval structures by fitting Gaussian mixture models.** To investigate whether the temporal structure of the exchange transitions and repeated transitions in F2F phases differed statistically, we applied Gaussian mixture models (GMMs) to the time interval data of each transition. The GMM attempts to estimate the number of mixtures and parameters (mean and variance), assuming that the data are generated as a mixture of multiple Gaussian distributions

consisting of different parameters. Previous studies on vocal exchange have shown that 1) the intervals follow logarithms and 2) the data are generated from two types of Gaussian distributions: a short time interval caused by the association of two vocalization transitions and a long time interval randomly caused by two independent vocalizations. Our aim of the GMM analysis here was to isolate only the distribution of "the time interval of associated two sequential vocalizations," which was our interest. By visually inspecting the bimodal distribution of the interval data, we fit the model with mixtures of the two components of Gaussian distributions to log-transformed data to estimate the parameters of the mixture models. Prior to log transformation, we excluded interval cases with negative values; that is, where the onset time of the subsequent call preceded the offset time of the preceding call, and there were only two such cases. Of these two components, the one estimating the shorter time interval ("short component," hereafter) was interpreted as the distribution of "the time interval of associated two sequential vocalizations" for each of exchange and repeated transitions. The other component estimating longer time intervals ("long component") was interpreted as the distribution of "the time interval between two independent vocalizations occurring at random." For the GMM fitting, we used `mclust` (v. 6.0.0) [49] in R (v. 4.2.3) [50].

After estimating the interval structure by fitting GMM, we performed a permutation test to statistically compare the interval between the exchange and repeated transitions as follows: First, we generated values from two normal distributions defined by the mean and variance of the intervals estimated by the GMM (two normal distributions for the exchange and repeat transitions), and calculated the difference between the generated values. This calculation is repeated 10,000 times to obtain a distribution of 10,000 values. We then calculated a 95th percentile interval, defined by the 2.5th and 97.5th percentiles of these values. By examining whether this interval contained zero, we determined whether a statistically significant difference existed between the two distributions. We used `rnorm()` function in R to generate data from the normal distribution.

We further investigated whether the presence of a conspecific influenced the temporal structure of intervals in repeated transitions. Specifically, we examined whether the intervals between pre- and post-transition phases differed, using GMM estimation and a permutation test. The results showed that the distributions of transition intervals in the PRE and POST phases were remarkably similar. The estimated means (± variance) of the short-latency component from the GMM were −0.75±0.01 and −0.074±0.334 for the PRE and POST phases, respectively. The 95th percentile range for the difference in means, obtained via permutation testing, ranged from −1.802 to 0.489 and included zero, indicating no significant difference between the two phases. Based on this result, we combined the PRE and POST phases into a single category, hereafter referred to as the "solo" phase. We then compared the interval distributions between the solo and face-to-face (F2F) phases using the same GMM estimation and permutation testing approach to assess whether they differed.

### Pseudo-turn-taking data by shuffling the vocal data

The short component observed in the exchange transition in the F2F phases could be interpreted as the reaction time to the preceding call in the vocal exchange, but we cannot rule out the possibility that this might have been caused by chance when two individuals vocalized independently at a certain interval of time. To investigate whether the distribution of the short component was generated by chance, we compared the pseudo-turn-taking data (generated with reference to [8]) with the actual exchange transition data. Pseudo-turn-taking data were created by shuffling and combining data from different sessions to simulate "virtual" vocal exchanges. Specifically, they were generated by pairing the onsets and offsets of the subject birds' calls with those of the facing birds, each extracted from separate sessions in the F2F condition. Thus, while the pairing remained consistent between the actual and pseudo-turn-taking data, the vocalizations of the subject birds and facing birds occurred independently in the pseudo-turn-taking condition. (Fig 2C).

### Statistical evaluation of the number of calls across experimental phases

To investigate whether the vocalizations of subject birds were influenced by those of the facing birds, we compared the number of vocalizations produced by the subject birds across the three phases (PRE, F2F, and POST) using generalized linear mixed models (GLMM) and model selection techniques. We fitted a GLMM (error distribution: Poisson distribution,

link function: log) with the number of vocalizations given in each session by the subject birds that were placed in the experimental space first (i.e., experienced all three experimental phases) as the explanatory variable, the three experimental phases as the fixed effect term, and the subject and trial as random intercepts. By comparing the Akaike information criterion (AIC), we estimated which model outperformed the null model in which the condition was removed from the fixed effects. GLMM fitting and AIC calculations were performed using the glmer function of lme4 (v. 1.1-34) [51] and the dredge function of MuMIn (v. 1.47.5) [52] in R.

### Ethical note

The experiments were reviewed and approved in advance by the Animal Experiment Ethics Committee of the University of Tokyo Graduate School of Arts and Sciences (#2022−1 and 2023−8).

### Results

#### Temporal distributions of call transitions

In the F2F phase, we recorded 5062 call transition sequences, of which 1712 were exchange transitions, and 3350 were repeat transitions. There were only two overlapping interactions in the exchange transitions. Fig 3 shows the distribution of the call offset onset intervals for exchange and repeated transitions. The mean with variance of the estimated short component of GMM are $-0.728 +/- 0.023$ and $-0.669 +/- 0.035$ for the exchange and repeated transitions, respectively (Fig 3A and 3B, Table 1; note that the values were log-transformed). This suggests that there was no statistical difference between the two transitions. Subsequently, using these estimated parameters, we performed a permutation test to determine the differences in values between the conditions. The calculated 95th percentile range of the value differences was from −0.533 to 0.403, which was not biased away from the 0.

When comparing the distribution of repeated transition of 'solo' and F2F phase by GMM, the estimated means (± variance) of the short-latency component were $-0.75 \pm 0.009$ and $-0.669 \pm 0.035$, respectively. The 95th percentile range

A                                                    B

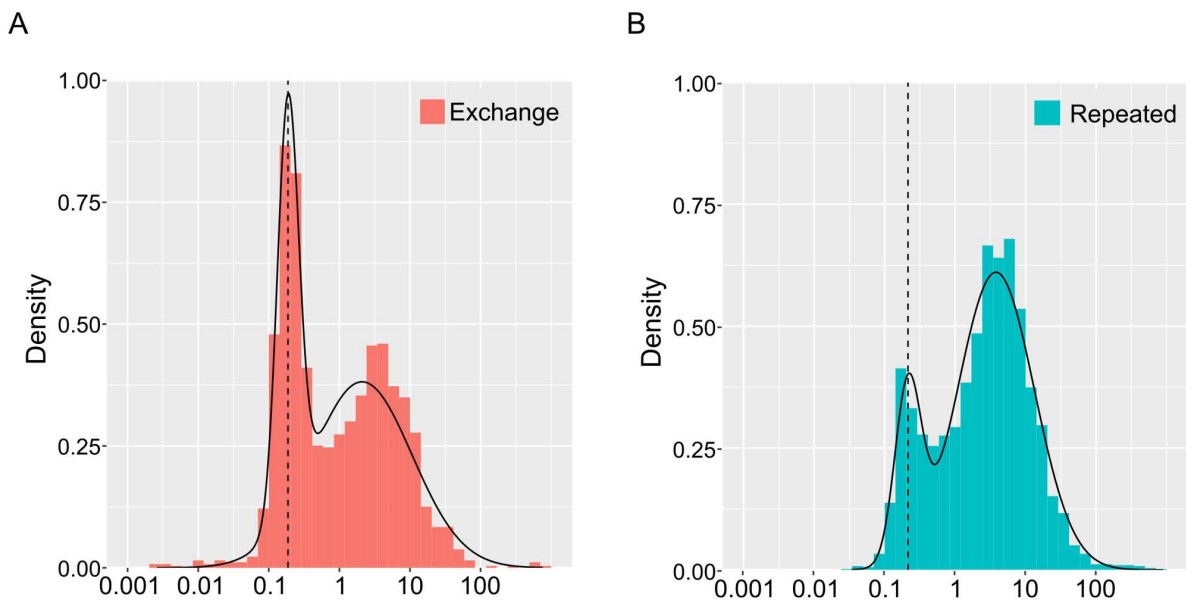

**Fig 3. Histogram and estimated probability density function curve (curve line) fitted by the two modal GMM for the exchange transition (A) and for repeated transitions (B).**

**Table 1. The parameter estimations of the GMMs fitting for the data of exchange transition and repeated transition.** The numbers in parentheses in the Mean, and Variance columns represent antilogarithmic values, converted from logarithms back to seconds to help understanding.

| Transition | Data size (N) | Components | Mean | | Variance | Mixture ratio |
|---|---|---|---|---|---|---|
| Exchange | 1710 | short | −0.728 | (0.187) | 0.023 | 0.322 |
| | | long | 0.321 | (2.094) | 0.502 | 0.678 |
| Repeated | 3350 | short | −0.669 | (0.214) | 0.035 | 0.167 |
| | | long | 0.578 | (3.784) | 0.296 | 0.833 |

for the difference in means, obtained via permutation testing, ranged from −0.329 to 0.505. These values include zero, indicating no significant difference between the two phases. These results suggest that Java sparrows showed vocal responses to facing individuals, as reflected in the shorter intervals during exchange transitions. However, the duration of repeated transitions was not affected by the mere presence of the other bird.

The dashed vertical lines represent the positions of the estimated mean parameter values in the short components of the GMMs. Note that although the data are log-transformed, the x-axis shows antilogarithmic values in seconds.

## Comparisons with pseudo-turn-taking data

In the pseudo-turn-taking data, we identified 948 exchange transitions and 4168 repeated transitions. There are three overlapping cases of exchange transition. Fig 4 shows the distribution of call offset-onset intervals for the exchange transitions of the pseudo-turn-taking data. The distribution of the pseudo-turn-taking data was uniform and monotonic, with no clear components such as the short component observed in the exchange transition. This strongly suggests that the

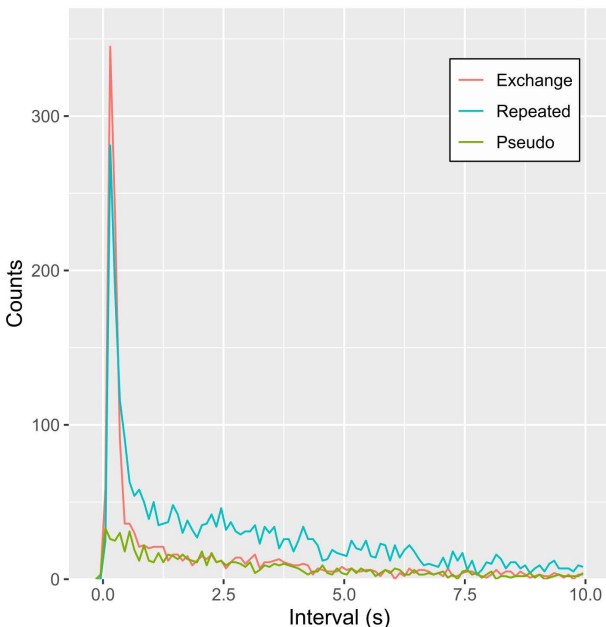

**Fig 4. Distributions of inter-call intervals of exchange transitions, repeated transitions and pseudo turn-taking data.** The red line represents the exchange transitions, the blue line represents the repeated transitions, and the green line represents the pseudo-turn-taking data. The length of the inter-call interval could be theoretically up to 900 seconds, but the X-axis here is limited to 10 seconds. This is because more than 80% of the inter-call interval was contained within 10 seconds (90.04% of exchange transition, 83.40% of repeated transition and 82.38% of pseudo transition).

short component identified in the exchange transitions originated from the "response" call of the paired individuals to the preceding call of the other individuals.

## Comparisons of the number of calls across experimental phases

We recorded 9608 calls during 60 sessions (900 min for each of the three conditions, totalling 2700 min). In the three experimental phases, the number of calls given by the birds increased in the F2F phase for five of the six subjects compared to the PRE and POST phases (Fig 5). The mean numbers of calls given by the birds first introduced into the experimental space in the pre-, F2F-, and post-phases were 34.5, 50.8, and 33.8, respectively. The GLMM analysis revealed that calls were most frequently made in the F2F phase (AIC: 4221.3; AIC: 4486.9; Fig 5).

## Discussion

In this study, we investigated call intervals given by two Java sparrows sequentially (exchange transitions) and those given by the same individual repeatedly (repeated transitions). The results showed that the time intervals of the short component did not differ between exchanged and repeated transitions in Java sparrows. In other words, the birds in our study repeatedly called at the same time intervals when calls from other individuals were most likely to occur during exchange transitions. This contrasts with the temporal regularity of turn-taking/vocal exchanges observed in social mammals, in which individuals remain silent during time intervals when the likelihood of vocal responses from other individuals is high, anticipating a response that leads to longer intervals in repeated transitions than in exchange transitions. However, as evidenced by the increase in the number of calls during the F2F phase, this does not imply that Java sparrows were calling

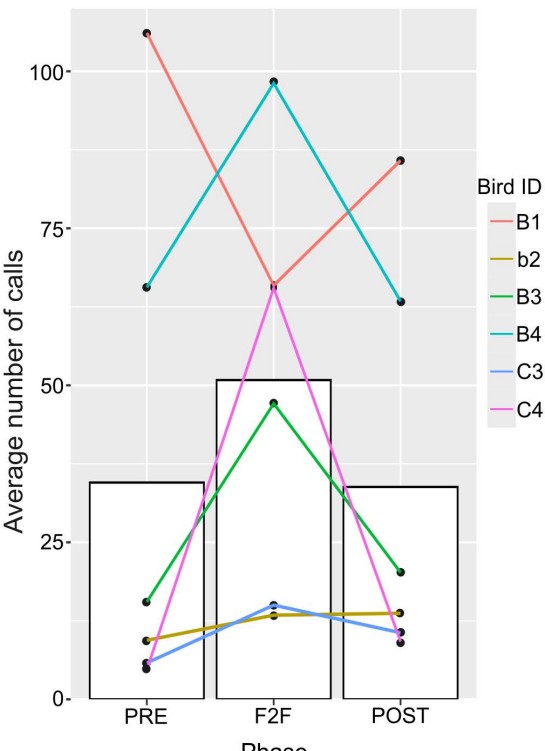

**Fig 5. Total number of calls given by six subjects in three experimental phases.** (A) The average number of calls in each phase. Each line indicates each bird. The female bird is indicated by the lowercase ID.

independently of other individuals. Rather, it is likely that Java sparrows responded contagiously to the calls of others, although they did not appear to anticipate a 'response' call.

In call exchange studies in birds, the stable and short time intervals between calls given by two individuals have been interpreted as vocal exchanges, similar to human turn-taking behavior [38]. In the present study, the intra-individual call intervals in the exchange transitions of Java sparrows were concentrated within a short temporal window. However, this result alone does not allow for the conclusion that the vocal exchange of Java sparrows shares characteristics similar to turn-taking in human conversations, as no significant difference in interval length was observed between repeated and exchange transitions. This indicates that, unlike the vocal turn-taking observed in socially communicative mammals, including humans, Java sparrows do not expect or wait for a response from other individuals.

Not predicting or expecting a response from the partner does not mean that Java sparrows give calls independently of the partner's call and do not communicate with each other. A comparison of the pseudo-turn-taking data and exchange transitions showed a short component only for the exchange transition. This indicates that they made calls in response to the preceding calls of the other individual. In addition, Java sparrows made more vocalizations in the F2F phase than in the standalone PRE and POST phases, although there were individual differences. This indicates that calls have an interactive communication function. In other words, Java sparrows vocally communicated with each other without predicting or anticipating a response from the facing individual, but calling contagiously in response to others' vocalizations. This is quite different from human conversation, in which turn-taking occurs at an appropriate time by predicting the vocalizations of the person to whom he or she is speaking [1].

Why do Java sparrows make contagious calls? We hypothesized that this would synchronize the movement of the flock. For birds that live in fluid flocks and can fly and move quickly, separation from a particular flock member is not as fatal as in mammals. Massive aggregation is an ecological characteristic of collective flocks of small birds, such as sparrows. In such large flocks, synchronizing the behavior among flock members is important for increasing the survival rate [53]. Behavioral synchronization has been widely reported in animals that live in a large mass of individuals: synchronization of vigilance or scanning behaviors to avoid predators [54,55], escape motion from predators within a flock [56], flight angles, and speeds between nearby individuals during flight [57]. Acoustic signals can function as cues for synchronized movements. Spatial coordination by calls has been experimentally reported in zebra finches flying in a wind tunnel [58]. In our study, we investigated only the vocal exchange between two individuals in separate cages. To test our hypothesis that contagious calling facilitates the coordination of flock movement, vocal exchange and behavioral synchronization among multiple individuals should be investigated in more natural, flock-like settings. The fact that they gave calls at similar intervals in both transitions suggests that many overlaps of calls by different individuals may have occurred, but only two cases were observed. Even in the pseudo-turn-taking data, the number of overlapping cases was as low as three. Therefore, it is plausible that the small number of overlapping cases was due to the short length of their calls (mean length of 24.7 msec). However, in Java sparrows, the call duration is so brief that even if the birds anticipate a response, there may be no need to avoid overlapping. This can be further clarified in future experiments using larger flocks. Specifically, in vocal exchanges involving more than two individuals, the chance of call overlap would be expected to increase simply due to the greater number of potential responders. If birds only respond reflexively without anticipating the timing of others' calls, such overlap is likely to occur more frequently. However, if individuals are capable of anticipating responses and coordinating their vocal timing, they may be able to avoid overlaps even in larger groups. Indeed, such overlap avoidance has been reported in zebra finches [37–40], where individuals adjust their timing to avoid calling simultaneously. Whether similar mechanisms operate in Java sparrows in more complex settings, such as those involving larger flocks, remains an open question.

In our study, Java sparrows were predominantly male, with a single female, and each individual was housed separately. Consequently, there were no individuals with affiliative relationships such as pairs. Given these conditions, it must be acknowledged that the social environment of the birds in our study is a significant simplification of their natural social lives.

In zebra finches housed in mixed-sex groups, vocal exchange between pairs has been shown to take priority, with vocal exchanges occurring at shorter intervals between pairs than in non-paired individuals [41]. Additionally, the call interval in mixed-sex groups is shorter than that in groups consisting only of females [59]. Given that, similar to zebra finches, pair bonds are fundamental in Java sparrows, it is reasonable to assume that the temporal regularity of vocal exchange may vary depending on social relationships. Although our study can be regarded as an investigation of the basic temporal regularity of vocal exchange in the absence of pair bonds, future research is essential to examine the influence of sex differences and social relationships on the temporal regularity of vocal exchange.

In general, when two individuals produce consecutive vocalizations in a fixed timeframe in non-human animals, this tends to be regarded as a vocal exchange with characteristics similar to turn-taking in human conversations. However, temporal regularities may differ between mammals and birds, as well as between species, such as zebra finches and Java sparrows, owing to variations in their ecological adaptations and social group structures. However, given the differences in results between zebra finches and Java sparrows, it is currently impossible to draw definitive conclusions on vocal turn-taking in birds. Furthermore, to enable meaningful comparisons between mammals and birds, research on call exchanges in birds should be conducted across a wide range of species. We hope that this study will encourage further research in the field of vocal exchange, ultimately shedding light on the similarities and differences between avian and mammalian vocal interactions, their functions, and their evolutionary significance.

## Acknowledgments

We are grateful to two anonymous reviewers for drastically improving our manuscript. We appreciate the administrative support provided by Riko Goto and Sana Kohmoto.

## Author contributions

**Conceptualization:** Sota Kikuchi, Noriko Kondo, Hiroki Koda.

**Data curation:** Sota Kikuchi, Noriko Kondo, Hiroki Koda.

**Formal analysis:** Sota Kikuchi.

**Funding acquisition:** Hiroki Koda.

**Investigation:** Sota Kikuchi, Noriko Kondo, Hiroki Koda.

**Methodology:** Sota Kikuchi, Noriko Kondo, Hiroki Koda.

**Project administration:** Hiroki Koda.

**Resources:** Sota Kikuchi, Noriko Kondo, Hiroki Koda.

**Software:** Sota Kikuchi, Hiroki Koda.

**Supervision:** Hiroki Koda.

**Validation:** Sota Kikuchi, Noriko Kondo, Hiroki Koda.

**Visualization:** Sota Kikuchi, Noriko Kondo.

**Writing – original draft:** Sota Kikuchi, Noriko Kondo, Hiroki Koda.

**Writing – review & editing:** Sota Kikuchi, Noriko Kondo, Hiroki Koda.

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
