## [Decision Letter · Decision Letter 0]

PONE-D-24-46388Contagious vocal reaction forms avian vocal exchange: an experimental study in Java sparrowsPLOS ONE

Dear Dr. Koda,

Thank you for submitting your manuscript to PLOS ONE. I apologize that the review process has been so long.After careful consideration, we feel that the manuscript has merit but does not fully meet PLOS ONE’s publication criteria as it currently stands. It will require substantial revision to be considered for publication, and will require re-review Therefore, we invite you to submit a revised version of the manuscript that fully addresses the points raised during the review process.

Both reviewers felt that the conclusions were too broad. The data are limited to one species, and the methodological issues too nuanced to conclude that avian vocal exchange operates by “a mechanism different from that of mammalian vocal exchange”. Please limit your interpretation and conclusion accordingly. 

Reviewer #1 raises various issues that must be addressed.Here is a partial list. In particular, clarify the question being tested and the exact social context in which calls were quantified. In addition, separate exchange transitions for alone vs. F2F, and clarify and account for the presence of a female bird.

Reviewer #2 suggests additional discussion of relevant literature and more conservative conclusions. Among other things, the reviewer requests clarification of how overlapping calls were assessed, and the need to compare these observations to chance occurrence.

Both reviewers list numerous specific concerns that must be addressed. Furthermore, a native English speaker should copy-edit the manuscript to correct typos and make the language clearer.

We look forward to receiving your revised manuscript.

Kind regards,

David S Vicario, Ph.D.

Academic Editor

PLOS ONE

Journal Requirements:

“This study was funded by the JSPS KAKENHI (B, #23K25168/22H03914; S, #23H05428) awarded to HK.”

“Japan Society for the Promotion of Science KAKENHI (B, #23K25168/22H03914; S, #23H05428) awarded to HK”

Reviewers' comments:

Reviewer's Responses to Questions

**Comments to the Author**

1. Is the manuscript technically sound, and do the data support the conclusions?

Reviewer #1: No

Reviewer #2: Partly

2. Has the statistical analysis been performed appropriately and rigorously? 

Reviewer #1: No

Reviewer #2: Yes

3. Have the authors made all data underlying the findings in their manuscript fully available?

Reviewer #1: Yes

Reviewer #2: Yes

4. Is the manuscript presented in an intelligible fashion and written in standard English?

Reviewer #1: No

Reviewer #2: Yes

5. Review Comments to the Author

Reviewer #1: Kikuchi et al. describe an experiment in which they examine call latency within and between Java sparrows. They claim that repeated call latency (when a bird continues calling without a response from another present bird) is shorter than similar types of intervals observed in mammals and other birds. While the use of comparative metrics is valuable, it’s unclear the extent to which these data are appropriate for their question and subsequently whether their conclusions are appropriate.

I appreciate the evaluation of how social dynamics within groups of species informs their question, but the broad generalization of the ‘fundamental difference of social structure in mammals and birds” (line 124) is quite sweeping and not testable using a single species of bird. While they compare their findings to published zebra finch data, it seems as though they’re own analyses differ significantly from those previous studies, so it is difficult to draw conclusions as to any similarities or differences between the species.

Also, it unclear as to what question(s) they’re attempting to test. Is it the calling pattern of single individual when in the presence/absence of another bird, or whether these calling patterns are fundamentally different from other bird/mammalian species? The authors should more clearly clarify the goal of their study. And, if they’re attempting to both questions, it’s not clear that they have sufficient evidence for the latter.

My main concern with the manuscript is that I’m uncertain as to what the researchers were measuring (all call intervals regardless of social context or only call intervals when in the presence of a conspecific (F2F). Several sentences are difficult to interpret and could be clarified – while these are just a few examples, there are others that may contribute to difficulty in understanding the mechanics of the manuscripts.

- Line 155: “They could usually hear and see each other”

- Line 160: “We designed the experimental space in which individual cages could be placed three-dimensionally”

- Line 189: “faced over the cage”

My understanding of the methodology is that Java sparrows called in two social contexts: alone (PRE and POST) and F2F. Total call number was greater in the F2F condition compared to the PRE/POST conditions. In addition, they measured the call interval in each of these contexts. In the F2F condition, the call interval was split into either an ‘exchange transition’ (a response to a partner bird) or a ‘repeated transition’ (the interval between two consecutive calls from the same individual).

By definition, ‘exchange transitions’ (te) can only occur in the F2F condition. However, ‘repeated transitions’ (tr) can occur in any social context. It appears the tr have all been grouped together. It would be helpful if the authors included an analysis of solely the F2F (not including any data from the PRE/POST conditions) phase. If that is what the authors did, it should be made clearer that only call intervals from the F2F phase were evaluated. Certainly, the birds must use visual and auditory cues to determine the presence/absence of another bird – the low tr interval could be explained by birds repeatedly calling in the absence of another bird in the PRE/POST conditions – the birds would not be expecting a response, so have no reason to avoid calling in a typical overlapping time interval. It would also be helpful to quantify the total percentage of returned calls – is the rate of return similar to what is observed in stranger zebra finches (Benichov et al., 2017)?

A comparison of tr in PRE/F2F/POST conditions would strengthen the manuscript to consider the effect of social context. Similarly, if the authors have video recordings of the birds, an account of whether the ‘partner’ bird is socially engaging the with the test bird, and thus, potentially more likely to respond to the test bird would also be a valuable addition. Lines 354-356 state that it’s the pseudo turn-taking data that lead to the strong suggestion that the short component identified in the exchange transition originated from the response call – the actual data from the two birds should be what leads to this suggestion because those data should and do exist.

I admit that I am not super familiar with the intricacies of GLMMs and GMMs. If the the GLMM considers the social context (alone vs. conspecific present), it would be helpful to more clearly illustrate and describe that process, along with various acronyms (e.g. AIC) that are used.

In the abstract, the authors state that the te latency is ~170ms. Nowhere in the manuscript do the authors report this result or explain its relevance (or adequately compare it to other mammalian or avian species). Similarly, they do not report a mean tr latency (for either solo vs. partner contexts) and how it compares to other mammalian or avian species (they report only that it is different – shorter).

Which bird was the female bird? While isolating a single bird’s performance in this experiment is likely not fruitful, including the female in the experiment leads to questions of whether the social dynamics between males/males and males/females is different.

I recommend several alterations to the figures to also strengthen the data presentation.

Figure 1: Include subscripts for the e and r in te and tr. Include a figure caption for 1B and perhaps an explanation for the predicted distributions.

Figure 2: Figure 2B – include a time scale

Figure 3: Indicate the female bird in Figure 3A. Figure 3B is unnecessary.

Figure 4-5: These are the critical figures. Figure 4 could be adjusted to account for ‘repeated transitions’ in the three contexts, to more carefully compare tr vs. te in F2F. (Also, is there a maximum latency in which a te or tr can occur? If so, what is the justification for those latencies?) Figures 4 and 5 could easily be combined.

In Figure 5 (according to the caption), a vertical dotted line represents the estimated values of the mean parameter in the short components). However, Table 5 (and lines 329-330) indicates negative values for those parameter estimates. It is unclear exactly what the dotted lines represent.

Lastly, the title of the manuscript is a bit odd considering that the term ‘contagious’ is not properly defined nor used until midway through the discussion.

Overall, as currently written, it is not possible to determine whether the authors conclusions and interpretations are justified due to the description and presentation of the data.

I applaud the authors for utilizing another avian species for a comparative study regarding vocal turn-taking. I also appreciate the pseudo turn-taking data as a control condition, and the inter-rater reliability metrics the authors provided regarding call classification, caller identification, and onset/offset times.

I selected, ‘No’ in response to ‘Is the manuscript presented in an intelligible fashion and written in standard English?’

While the manuscript is written in very readable English, there are enough typographical and language errors that lead to reader confusion that I suggest the authors address in future revisions.

Reviewer #2: Kikuchi et al investigate the temporal patterns of vocal exchanges in Java sparrows, with the aim of comparing communication strategies of birds to those reported in mammals. Specifically, the study aims to tests if, during vocal interactions, Java sparrows wait for a partner’s response before they produce their next call (a pattern associated with mammalian vocal turn-taking). The authors recorded face-to-face call interactions in randomized pairs of birds and analyze the temporal patterns of calls. They report that the latencies of call responses do not differ from the intervals between repeated calls of individuals. This is interpreted as evidence that birds do not adjust the timing of their vocalizations by anticipating or waiting for the responsive calls of others. Rather, in contrast to mammals, their call responses are “contagious” or socially triggered by the vocalizations of other individuals. The authors assert that these purely responsive vocal patterns correspond to the relatively fluid composition of avian social groups and aid in synchronizing flock behavior.

The study’s methods, analyses, and results are sound and generally presented in a clear, convincing manner. Some of the authors’ conclusions, however, extend beyond what was tested with their experimental paradigm, particularly in their generalization across bird species and social contexts. Nevertheless, the work presents valuable data and comparative insights from a species for which little is known regarding interactive vocal behavior. Most concerns described below can be addressed by reducing the generality of some claims and providing additional information, for example regarding the occurrence of overlaps and prevalence of repeated calls.

Comments:

1) Lines 20-22, 100-101: The authors state that no study has investigated the intervals between repeated calls of individual birds during interactions. However, intra-individual call patterns have indeed been examined in groups of zebra finches (e.g. Beckers & Gahr 2010 and Stowell et al 2016). The latter study specifically shows that temporal distributions of “self-self” call patterns (i.e. repeated transitions) differ across sexes and call types. And with particular relevance for the current study, patterns of intra-individual call timing were found to differ from responses to others (exchange transitions). “Self-partner” response patterns were found to further differentiate as breeding pairs were established. In addition to identifying excitatory influences of calls from self and others, their analyses also characterized patterns of call suppression, includes strong self-suppression, which would suggest mechanisms more complex than solely excitatory contagion.

Kikuchi et al do acknowledge that in other songbirds, familiarity and pair-bonding can affect the temporal patterns of call responses (e.g. faster, higher-likelihood responses to mates). However, since their experimental design is based on randomized, relatively brief pairings in Java sparrows, and does not take factors such as sex, call type, or affiliative pair bonding into account, the authors should be a bit more conservative with their resulting claims and predictions, rather than generalizing across the diversity of birds and social contexts: “We tentatively predict that avian vocal exchange is established by a mechanism different from that of mammalian vocal exchange in social groups - i.e. birds do not predict and wait in anticipation of vocalizations from other individuals.” (Lines 383-386).

2) The authors state that instances of call overlap are extremely rare (only 2/1712 exchange transitions), which is surprising if these birds respond to and repeat their calls with the same stereotyped intervals. It is suggested that the rarity of overlaps could be attributed to the short duration of calls. However, this finding is relevant to their claims that birds don’t anticipate the calls of others and should be investigated further. The authors should compare the experimentally observed instances of overlap to overlap rates that would be expected by chance (based on shuffled or simulated data with the same call rates). If short call durations can indeed explain low instances of call overlaps, this may suggest that Java sparrows simply lack the need for anticipatory mechanisms seen in other species with longer calls at higher risk of call overlap/signal masking. If overlaps occur less than expected by chance, other possible explanations are auditory suppression of repeated calls by responsive calls (e.g. Larter & Ryan 2024; Norton et al 2022) or the general self-suppression of repeat calls, which could indeed be indicative of an expectation of a vocal response or the result of differential usage of repeated calls across contexts. With the current dataset, the authors could at least compare the relative frequencies of repeated calls (short component) of subject individuals in the F2F vs Pre & Post phases.

3) In each experimental pairing, the subject individual was given a 15 minute acclimation period (PRE phase), whereas the facing bird was introduced immediately prior to the test. Relocating birds has been shown to affect vocalization rates (Zai et al 2025). Did transporting birds have effects on call rates? Specifically, were there any asymmetries in calling behavior when birds were in the subject vs. facing role/more vs. less acclimated?

Additional Minor Comments:

Lines 40-41, 46-47: There is evidence that the inter-speaker intervals during conversational turn-taking can be influenced by linguistic information/the speech planning network (e.g. Castellucci et al 2022).

Line 257: AIC should be defined as Akaike Information Criterion at first use.

Line 353: “Monotonic” rather than “Monotonous”.

Lines 382-383: It is not clear how the absence of “temporal regularity of vocal exchanges” has been determined. The authors previously defined temporal regularity as “stable time interval between vocalizations of two individuals and significant avoidance of vocal overlap” (lines 54-55). Is the “short component” distribution of exchange transitions not stable? Also, as mentioned above, the occurrence of overlaps is reported to be very low, but not assessed in terms of its significance. Greater consistency in the use of terms would be helpful here.

Line 401: “Why *do Java sparrows call contagiously?”

References:

Beckers, G. J., & Gahr, M. (2010). Neural processing of short-term recurrence in songbird vocal communication. PLoS One, 5(6), e11129.

Stowell, D., Gill, L., & Clayton, D. (2016). Detailed temporal structure of communication networks in groups of songbirds. Journal of the Royal Society Interface, 13(119), 20160296.

Larter L. C. and Ryan M. J. (2024) Sensory-motor tuning allows generic features of conspecific acoustic scenes to guide rapid, adaptive, call-timing responses in túngara frogs Proc. R. Soc. B.29120240992

P. Norton, J.I. Benichov, M. Pexirra, S. Schreiber, & D. Vallentin, (2022) A feedforward inhibitory premotor circuit for auditory–vocal interactions in zebra finches, Proc. Natl. Acad. Sci. U.S.A. 119 (23) e2118448119.

Zai A.T., Rodrigues D.I., Stepien A.E., Lorenz C., Giret N., Adam I., et al. (2025) Familiarity of an environment prevents song suppression in isolated zebra finches. PLoS ONE 20(1): e0307126.

Castellucci, G.A., Kovach, C.K., Howard, M.A. et al. (2022). A speech planning network for interactive language use. Nature. 602, 117–122

6. PLOS authors have the option to publish the peer review history of their article (what does this mean? ). If published, this will include your full peer review and any attached files.

**Do you want your identity to be public for this peer review?** For information about this choice, including consent withdrawal, please see our Privacy Policy .

Reviewer #1: No

Reviewer #2: No

---

## [Author Response · Author response to Decision Letter 1]

9 Apr 2025

Reply to reviewers and editors

We sincerely appreciate your evaluation and the insightful reviews of our manuscript submitted to PLoS ONE. We are particularly grateful for your recognition of our efforts in clarifying the temporal patterns of vocal exchange in Java sparrows. We also appreciate your positive feedback regarding the clarity of our experimental methods and analyses.

As turn-taking and vocal exchange are fundamental features of mammal communication, including human conversation, it has been assumed in previous research that similar patterns would be observed in birds as well. However, our study, which demonstrates that such patterns are not necessarily present in Java sparrows, provides valuable insight into this area. We believe our findings add significant value to the existing body of research.

That said, as pointed out by the editor and two reviewers, our original manuscript overgeneralized the findings from a single sparrow species to "avian" characteristics. Based on the valuable comments we received, we have made substantial revisions to the manuscript. We believe that, with these extensive revisions, the manuscript is now well-suited for publication in PLoS ONE.

Below, we provide our detailed responses to each comment received. The numbers assigned to the reviewer comments (e.g., R1-1) correspond to the highlighted sections in the marked-up copy of the revised manuscript.

We sincerely hope to address the concerns you raised regarding the previous version of the manuscript.

Hiroki Koda, Noriko Kondo, Sota Kikuchi

Reviewer #1:

R1-1

Kikuchi et al. describe an experiment in which they examine call latency within and between Java sparrows. They claim that repeated call latency (when a bird continues calling without a response from another present bird) is shorter than similar types of intervals observed in mammals and other birds. While the use of comparative metrics is valuable, it’s unclear the extent to which these data are appropriate for their question and subsequently whether their conclusions are appropriate.

I appreciate the evaluation of how social dynamics within groups of species informs their question, but the broad generalization of the ‘fundamental difference of social structure in mammals and birds” (line 124) is quite sweeping and not testable using a single species of bird. While they compare their findings to published zebra finch data, it seems as though they’re own analyses differ significantly from those previous studies, so it is difficult to draw conclusions as to any similarities or differences between the species.

You are absolutely correct. It was inappropriate to generalize our findings to all avian species based solely on a single species, the Java sparrow. In response, we have revised the relevant sections, including the title, to mitigate the generalization and refrain from making conclusions that apply to all avian species. The specific revisions are as follows:

Title: Temporal regularities of vocal exchange in Java sparrows

L. 114-120

Regarding the need to maintain group cohesiveness, mammals and birds are expected to differ from other individuals in their anticipated responses. Specifically, mammals are more likely to expect and anticipate responses, whereas birds may not share the same expectations. However, research examining the temporal characteristics of exchanges and repeated transitions in this field has been limited to zebra finches. Therefore, to conduct a comprehensive comparison of vocal turn-taking between these two taxa, it is necessary to expand our knowledge of vocal turn-taking in birds by investigating species other than zebra finches.

R1-2

Also, it unclear as to what question(s) they’re attempting to test. Is it the calling pattern of single individual when in the presence/absence of another bird, or whether these calling patterns are fundamentally different from other bird/mammalian species? The authors should more clearly clarify the goal of their study. And, if they’re attempting to both questions, it’s not clear that they have sufficient evidence for the latter.

My main concern with the manuscript is that I’m uncertain as to what the researchers were measuring (all call intervals regardless of social context or only call intervals when in the presence of a conspecific (F2F). Several sentences are difficult to interpret and could be clarified – while these are just a few examples, there are others that may contribute to difficulty in understanding the mechanics of the manuscripts.

Line 155: “They could usually hear and see each other”

Line 160: “We designed the experimental space in which individual cages could be placed three-dimensionally”

Line 189: “faced over the cage”

Our goal was to determine whether the general trends observed in previous studies on turn-taking in mammals, as shown in Fig. 1, could also be observed in Java sparrows. The answer to your question, "Is it the calling pattern of a single individual when in the presence/absence of another bird, or whether these calling patterns are fundamentally different from other bird/mammalian species?" would be the latter.

　The reason why our research objectives, results, and discussion may not have been properly understood is related to the next comment—namely, that our analysis target was not sufficiently described. We kindly ask you to consider the following comment in conjunction with this one.

　We are confident that, after doing so, you will recognize that the experimental results in this study on Java sparrows differ from the turn-taking characteristics illustrated in Fig. 1.

　Regarding the points you raised as examples of ambiguity in our experiment, we have made the following revisions in the revised version.

L.142-143

All individuals were housed separately in identical cages (38 cm high × 31 cm wide × 25 cm deep) within the same housing area, where they could hear and see each other.

L.151-153

Within the chamber, we designed an experimental space that allowed individual cages to be placed opposite to each other, facing one another (Fig. 2A).

L.178-179

In the F2F phase, the other individual (facing individual) was placed opposite the subject bird facing it.

R1-3

My understanding of the methodology is that Java sparrows called in two social contexts: alone (PRE and POST) and F2F. Total call number was greater in the F2F condition compared to the PRE/POST conditions. In addition, they measured the call interval in each of these contexts. In the F2F condition, the call interval was split into either an ‘exchange transition’ (a response to a partner bird) or a ‘repeated transition’ (the interval between two consecutive calls from the same individual).

By definition, ‘exchange transitions’ (te) can only occur in the F2F condition. However, ‘repeated transitions’ (tr) can occur in any social context. It appears the tr have all been grouped together. It would be helpful if the authors included an analysis of solely the F2F (not including any data from the PRE/POST conditions) phase. If that is what the authors did, it should be made clearer that only call intervals from the F2F phase were evaluated. Certainly, the birds must use visual and auditory cues to determine the presence/absence of another bird – the low tr interval could be explained by birds repeatedly calling in the absence of another bird in the PRE/POST conditions – the birds would not be expecting a response, so have no reason to avoid calling in a typical overlapping time interval. It would also be helpful to quantify the total percentage of returned calls – is the rate of return similar to what is observed in stranger zebra finches (Benichov et al., 2017)?

We apologize for the insufficient description of our methods and the structure of the manuscript, which may have led to a miscommunication of our experimental details. In the revised manuscript, we have made several modifications to ensure that our experimental procedures are clearly conveyed. The 'Statistical evaluation of the number of calls across experimental phases' section was initially placed at the beginning of the Statistics section in the original manuscript. This arrangement could have led to a misunderstanding, as it might have appeared that we were comparing call intervals between the three phases. However, the comparison between the three phases was conducted to examine whether the call frequency in Java sparrows was influenced by other individuals. Therefore, in the revised manuscript, we have moved this section to the end of the Statistics section (L.286-297 in the revised manuscript).

　Additionally, we have added clarifications to ensure the purpose of this analysis is clear. Furthermore, by appropriately placing the term ‘F2F phases’ in the manuscript, we have made the content of the analysis clearer. Specifically, the following changes have been made:

L.230

For the analysis of vocal exchange, we only used recordings of the F2F phases.

L.241-243

To investigate whether the temporal structure of the exchange transitions and repeated transitions in F2F phases differed statistically, we applied Gaussian mixture models (GMMs) to the time interval data of each transition.

A comparison of tr in PRE/F2F/POST conditions would strengthen the manuscript to consider the effect of social context. Similarly, if the authors have video recordings of the birds, an account of whether the ‘partner’ bird is socially engaging the with the test bird, and thus, potentially more likely to respond to the test bird would also be a valuable addition. Lines 354-356 state that it’s the pseudo turn-taking data that lead to the strong suggestion that the short component identified in the exchange transition originated from the response call – the actual data from the two birds should be what leads to this suggestion because those data should and do exist.

In this experiment, we focused exclusively on the F2F phase, as our interest was in vocal exchanges between two birds, and did not compare it to the PRE and POST phases. We appreciate your valuable comments on videos. However, unfortunately, we do not have any videos available for analysis in this study. However, we sincerely appreciate your comment and will ensure to collect videos for more detailed analysis in future experiments.

R1-4

I admit that I am not super familiar with the intricacies of GLMMs and GMMs. If the the GLMM considers the social context (alone vs. conspecific present), it would be helpful to more clearly illustrate and describe that process, along with various acronyms (e.g. AIC) that are used.

In the revised manuscript, we have clarified that a comparison of the three phases was conducted in the GLMM description. As per your suggestion, we have also spelled out AIC instead of using the abbreviation.

L.293-295

By comparing the Akaike information criterion (AIC), we estimated which model outperformed the null model in which the condition was removed from the fixed effects.

R1-5

In the abstract, the authors state that the te latency is ~170ms. Nowhere in the manuscript do the authors report this result or explain its relevance (or adequately compare it to other mammalian or avian species). Similarly, they do not report a mean tr latency (for either solo vs. partner contexts) and how it compares to other mammalian or avian species (they report only that it is different – shorter).

As you pointed out, the value of 170 msec was not mentioned in the Results section of the original manuscript. We recognized that this value does not play a significant role in our findings, and as such, we have removed it from both the Abstract and the Discussion. Specifically, the following changes have been made:

L.21-22

The results revealed that they vocalized at very short intervals following the calls of the other individual.

L.380-381

A comparison of the pseudo turn-taking data and exchange transitions showed a short component only for the exchange transition.

R1-6

Which bird was the female bird? While isolating a single bird’s performance in this experiment is likely not fruitful, including the female in the experiment leads to questions of whether the social dynamics between males/males and males/females is different.

We completely agree with your comment that the inclusion of only one female in our experiment could have various potential effects on the results. Considering that our original manuscript lacked sufficient discussion on this matter, we have added the following explanation in the Discussion section of the revised manuscript:

L.415-427

In our study, Java sparrows were predominantly male, with a single female, and each individual was housed separately. Consequently, there were no individuals with affiliative relationships such as pairs. Given these conditions, it must be acknowledged that the social environment of the birds in our study is a significant simplification of their natural social lives. In zebra finches housed in mixed-sex groups, vocal exchange between pairs has been shown to take priority, with vocal exchanges occurring at shorter intervals between pairs than in non-paired individuals [41]. Additionally, the call interval in mixed-sex groups is shorter than that in groups consisting only of females [59]. Given that, similar to zebra finches, pair bonds are fundamental in Java sparrows, it is reasonable to assume that the temporal regularity of vocal exchange may vary depending on social relationships. Although our study can be regarded as an investigation of the basic temporal regularity of vocal exchange in the absence of pair bonds, future research is essential to examine the influence of sex differences and social relationships on the temporal regularity of vocal exchange.

I recommend several alterations to the figures to also strengthen the data presentation.

We appreciate your comments. We have changed the figures as follows as per your comments. Please note that the figure numbers (Fig 3 and Fig 5) have changed due to the alteration in the order of appearance in the revised manuscript.

R1-7

Figure 1: Include subscripts for the e and r in te and tr. Include a figure caption for 1B and perhaps an explanation for the predicted distributions.

We included subscripts for the te and tr in the Figure1A (“Inter-call interval of exchange transition (te)” and “Inter-call interval of repeated transition (tr)”) as has been suggested. We added an explanation of Figure 1B in the caption of Figure 1. We also changed “animal” in the first sentence to “social mammals” to be consistent with the title of Figure 1.

L.61-67

Fig 1. Conceptual representation of the mechanism underlying the emergence of temporal regularity in vocal exchange/turn-taking of various social mammals.

When social mammals are engaged in a reciprocal vocal exchange (A, top), the individual expects a “response” from the other individual and therefore waits for a response. After the expected response time had elapsed, the individual repeated the vocalization (A, bottom). Fig 1B illustrates the inter-call intervals of vocal exchange in mammals. The inter-call interval of the repeated transitiontr is longer than that of the exchange transition te (A, B).

Figure 2: Figure 2B – include a time scale

We included a time scale bar to Figure 2B.

R1-8

Figure 3: Indicate the female bird in Figure 3A. Figure 3B is unnecessary.

We showed the female bird ID in lowercase letter (b2) in Fig 5 (Fig 3 in the original manuscript). Accordingly, we have changed the caption of Fig 5 as follows.　

L.350-352

Fig 5. Total number of calls given by six subjects in three experimental phases.

(A) The average number of calls in each phase. Each line indicates each bird. The female bird is indicated by the lowercase ID.

We have removed Figure 3B of the original manuscript as per your suggestion.

R1-9

Figure 4-5: These are the critical figures. Figure 4 could be adjusted to account for ‘repeated transitions’ in the three contexts, to more carefully compare tr vs. te in F2F. (Also, is there a maximum latency in which a te or tr can occur? If so, what is the justification for those latencies?)

We apologize that we have not clearly explained our experiments in the origi

---

## [Decision Letter · Decision Letter 1]

PONE-D-24-46388R1Temporal regularities of vocal exchange in Java sparrowsPLOS ONE

Dear Dr. Koda,

Thank you for submitting your manuscript to PLOS ONE. After careful consideration, we feel that it has merit but does not fully meet PLOS ONE’s publication criteria as it currently stands. Therefore, we invite you to submit a revised version of the manuscript that addresses the points raised during the review process. **I apologize for the long turn-around time for your revised manuscript. I felt it was important to get comments from both of the original reviewers and might take longer if new reviewers were engaged. ****Both Reviewers feel the paper has been significantly improved. However, there are some further concerns. Please respond to all the specific comments, especially those of Reviewer #1.** Please submit your revised manuscript by Jul 12 2025 11:59PM. If you will need more time than this to complete your revisions, please reply to this message or contact the journal office at plosone@plos.org . Please include the following items when submitting your revised manuscript:

We look forward to receiving your revised manuscript.

Kind regards,

David S Vicario, Ph.D.

Academic Editor

PLOS ONE

Reviewers' comments:

Reviewer's Responses to Questions

**Comments to the Author**

1. If the authors have adequately addressed your comments raised in a previous round of review and you feel that this manuscript is now acceptable for publication, you may indicate that here to bypass the “Comments to the Author” section, enter your conflict of interest statement in the “Confidential to Editor” section, and submit your "Accept" recommendation.

Reviewer #1: (No Response)

Reviewer #2: All comments have been addressed

2. Is the manuscript technically sound, and do the data support the conclusions?

Reviewer #1: Yes

Reviewer #2: Yes

3. Has the statistical analysis been performed appropriately and rigorously? 

Reviewer #1: Yes

Reviewer #2: Yes

4. Have the authors made all data underlying the findings in their manuscript fully available?

Reviewer #1: Yes

Reviewer #2: Yes

5. Is the manuscript presented in an intelligible fashion and written in standard English?

Reviewer #1: Yes

Reviewer #2: Yes

6. Review Comments to the Author

**Reviewer #1:**  The manuscript by Kikuchi et al is greatly improved following reviewer feedback. I appreciate the time the authors spent revising the manuscript to make the methodology and results clearer and the data interpretation more focused.

The generalization between birds and mammals, both of which are large and varied taxonomic groups, has been mostly eliminated. Ideally, the lingering references to the larger groups (e.g. line 25: “This contrasts with the temporal characteristics of vocal exchanges observed in mammals”), should be focused further to not suggest that the scientific community is certain that all mammals (or birds) share this trait – this is not something we can/cannot know. While the introduction does a fine job of providing more nuanced explanations of that sentence in particular, refraining from overgeneralizing should be avoided.

There is an additional analysis that may add to the help clarify why no differences were observed between tr and te transitions. Do tr intervals change based on social context? In other words, are the tr intervals longer or shorter when the bird has no partner bird to exchange calls with? Comparing the tr interval across the 3 phases of the experiment (PRE, F2F, POST) may provide useful insight into the utility of that transition based on whether the bird is alone or not. If it is the same across all three contexts, it might indicate that the birds are not using a call to communicate with a partner birds. However, if the tr is longer in the F2F phase compared to the other two that could suggest that they are awaiting a potential response. Similarly, if the POST tr interval is different from PRE or F2F, that might also suggest the role of the call in an attempt to contact conspecifics.

This suggested analysis (or something similar) might also help address the gap that the authors identify on lines 92-95 (“…while studies on the intervals between repeated transitions remain limited”). The current manuscript, as written, focuses more on the differences (or lack thereof) between repeated and exchanged transitions. An analysis focused solely on the repeated transitions would better address this gap.

To further strengthen the argument presented in the introduction, I encourage the authors to revisit the paragraphs on pages 5-6 (lines 96-121). I wonder if swapping the order of those two paragraphs might be helpful, particularly because lines 116-117 (“Specifically, mammals are more likely to expect and anticipate responses, whereas birds may not share the same expectations”) beg a “why?” even though it has been addressed to some extent in the previous paragraph. (Also, moving this sentence might be helpful to contextualize which birds and mammals have been studied to avoid the overgeneralization to all birds and/or mammals.

Given that I had difficulty interpreting Table 1/Figure 3 initially, the inclusion of the antilogarithmic values is extremely helpful. Please include the x-axis legend on Figure 3, specifying that you are plotting the antilogarithmic values. Additionally, I wonder if reorganizing Table 1 might be helpful by having the columns reflect the short and long components with the mean ± variance in the cells. Then you could more easily present the log transformed and anti-log values for each component.

Overall, modifications to the manuscript have helped showcase how calling behavior in java sparrows differs from those of other birds/mammals to better explain the utility of contagious calling and the value of comparative studies to evaluate how and why different groups of animals generate and modify vocal turn-taking.

Minor suggestions/observations:

1) Line 53, clarify ‘it’

2) Line 111-112, embed/modify the ‘birds fly much faster than mammals’ – there aren’t many mammalian species that fly, so referring to their movement as ‘travel’ may be more appropriate.

3) Consider combining the sentences on lines 125-127 for readability.

4) Line 178 – replace ‘enhance’ with ‘promote’ or something similar

5) Line 179 – replace ‘incubated’ – “The duration of the F2F phase was 15 minutes.”

6) Line 181-2: replace with “During the POST phase, facing individuals were removed from the soundproof room…”

7) Line 390 – use of the phrase, “affected by the vocal behavior of other individuals” doesn’t seem appropriate given that if you’d found differences between the transitions you would have been able to write this exact phrase. Consider clarifying.

8) Line 402 – modify to “To more appropriately test our hypothesis we will need to investigate vocal exchange …”

**Reviewer #2:**  The authors have sufficiently addressed my previous comments and the revised manuscript is considerably improved.

I would just like to point out the following minor phrasing issues, which can be easily edited prior to publication:

Lines 368-369: “Instead, they called in synchrony…” can be too easily misinterpreted as “called simultaneously”, which they apparently do not do in this paradigm, given the low instances of overlap. This new sentence can be removed.

Lines 402-404 “To test our hypothesis, we had to investigate the vocal exchanges and behavioral synchronization among more than three individuals in more natural, flock-like settings.”

- This was not done here. Perhaps authors meant “we would have to investigate…”.

-Also “our hypothesis” is a bit confusing here, given that the previous sentence refers to “our study”. Text should specify “To test our hypothesis regarding the synchronization of flocks, we would have to investigate…” (or similar).

Lines 414-415: It is not clear to me that the ability to avoid overlapping with a specific partner or in a playback experiment would necessarily mean that call overlaps would not increase in the presence of more callers. This new sentence can be removed.

7. PLOS authors have the option to publish the peer review history of their article (what does this mean? ). If published, this will include your full peer review and any attached files.

**Do you want your identity to be public for this peer review?** For information about this choice, including consent withdrawal, please see our Privacy Policy .

Reviewer #1: No

Reviewer #2: No

---

## [Author Response · Author response to Decision Letter 2]

3 Jun 2025

Reply to reviewers and editors

Reviewer #1:

R1-1

The manuscript by Kikuchi et al is greatly improved following reviewer feedback. I appreciate the time the authors spent revising the manuscript to make the methodology and results clearer and the data interpretation more focused.

The generalization between birds and mammals, both of which are large and varied taxonomic groups, has been mostly eliminated. Ideally, the lingering references to the larger groups (e.g. line 25: “This contrasts with the temporal characteristics of vocal exchanges observed in mammals”), should be focused further to not suggest that the scientific community is certain that all mammals (or birds) share this trait – this is not something we can/cannot know. While the introduction does a fine job of providing more nuanced explanations of that sentence in particular, refraining from overgeneralizing should be avoided.

We are grateful for the reviewer’s insightful comment. To better emphasize the point that these findings pertain to mammals studied to date, we have revised the sentence in the manuscript as follows:

L. 25

This contrasts with the temporal characteristics of vocal exchanges observed in social mammals that have been studied to date.

R1-2

There is an additional analysis that may add to the help clarify why no differences were observed between tr and te transitions. Do tr intervals change based on social context? In other words, are the tr intervals longer or shorter when the bird has no partner bird to exchange calls with? Comparing the tr interval across the 3 phases of the experiment (PRE, F2F, POST) may provide useful insight into the utility of that transition based on whether the bird is alone or not. If it is the same across all three contexts, it might indicate that the birds are not using a call to communicate with a partner birds. However, if the tr is longer in the F2F phase compared to the other two that could suggest that they are awaiting a potential response. Similarly, if the POST tr interval is different from PRE or F2F, that might also suggest the role of the call in an attempt to contact conspecifics.

This suggested analysis (or something similar) might also help address the gap that the authors identify on lines 92-95 (“…while studies on the intervals between repeated transitions remain limited”). The current manuscript, as written, focuses more on the differences (or lack thereof) between repeated and exchanged transitions. An analysis focused solely on the repeated transitions would better address this gap.

We sincerely appreciate the reviewer’s insightful comments. In response to the suggestions, we have conducted additional analyses accordingly. As a result, we have added new sections to both the Methods and Results in the revised manuscript. Since no difference was found in repeated transitions based on the presence or absence of a partner, we have refrained from discussing this point in depth. We hope this approach is understandable and satisfactory to the reviewer.

L. 278-289

We further investigated whether the presence of a conspecific influenced the temporal structure of intervals in repeated transitions. Specifically, we examined whether the intervals between pre- and post-transition phases differed, using GMM estimation and a permutation test. The results showed that the distributions of transition intervals in the PRE and POST phases were remarkably similar. The estimated means (± variance) of the short-latency component from the GMM were –0.75 ± 0.01 and –0.074 ± 0.334 for the PRE and POST phases, respectively. The 95th percentile range for the difference in means, obtained via permutation testing, ranged from –1.802 to 0.489 and included zero, indicating no significant difference between the two phases. Based on this result, we combined the PRE and POST phases into a single category, hereafter referred to as the "solo" phase. We then compared the interval distributions between the solo and face-to-face (F2F) phases using the same GMM estimation and permutation testing approach to assess whether they differed.

and also

L. 334-341

When comparing the distribution of repeated transition of ‘solo’ and F2F phase by GMM, the estimated means (± variance) of the short-latency component were -0.75 ± 0.009 and -0.669 ± 0.035, respectively. The 95th percentile range for the difference in means, obtained via permutation testing, ranged from -0.329 to 0.505. These values include zero, indicating no significant difference between the two phases. These results suggest that Java sparrows showed vocal responses to facing individuals, as reflected in the shorter intervals during exchange transitions. However, the duration of repeated transitions was not affected by the mere presence of the other bird.

R1-3

To further strengthen the argument presented in the introduction, I encourage the authors to revisit the paragraphs on pages 5-6 (lines 96-121). I wonder if swapping the order of those two paragraphs might be helpful, particularly because lines 116-117 (“Specifically, mammals are more likely to expect and anticipate responses, whereas birds may not share the same expectations”) beg a “why?” even though it has been addressed to some extent in the previous paragraph. (Also, moving this sentence might be helpful to contextualize which birds and mammals have been studied to avoid the overgeneralization to all birds and/or mammals.

We sincerely appreciate the reviewer’s insightful comments. In accordance with the suggestion, we have rearranged the paragraphs and revised the expressions to improve the logical flow of the argument. We have also carefully modified the phrasing to avoid the unintended impression that “mammals” refers to all mammalian species. The corresponding revisions have been made in the manuscript as follows.

L. 97-127

When considering the role of vocal turn-taking in animal societies, the socio-ecological structure of the species in question should be taken into account. Social mammals form stable groups with fixed membership even outside the breeding season, often including unrelated individuals. In such groups, vocal turn-taking helps maintain group cohesion by enabling individuals to signal their location and avoid separation through reciprocal vocal responses [42]. Birds, on the other hand, also form social flocks, but these tend to be more fluid and less stable than mammalian social structures. Within bird flocks, the most salient social units are often pair-mates, family members, or kin. For example, species such as Greylag geese (Anser anser) form matrilineal alliances [43], yet they do not necessarily maintain exclusive territories with fixed group membership, as is often observed in mammal groups. Given these socio-ecological differences, along with the fact that birds in general are capable of traveling much faster than mammals due to their ability to fly, temporary spatial separation from flock members likely poses a lower risk for birds than for group-forming social mammals. Accordingly, vocal turn-taking with an expectation of a response appears to be less essential for birds than for social mammals as a mechanism for maintaining proximity and group cohesion.

As is evident from recent zebra finch studies, however, the vocal exchange of calls in birds has been increasingly examined using a framework similar to that applied to primates. Previous studies have shown that humans, non-human primates, mammals, and birds share common vocal exchange features—specifically, short inter-individual intervals and longer intra-individual intervals, both of which help to avoid call overlap. Due to these shared characteristics, it has been hypothesized that vocal exchange may serve comparable functions across taxa. Nevertheless, as noted above, fundamental socioecological differences between social mammals and birds may shape their communicative strategies in distinct ways. Importantly, most avian vocal exchange research has focused on zebra finches, a key model species in songbird studies. These studies, which involved controlled face-to-face experiments, demonstrated significant avoidance of vocal overlap and active adjustments in call timing [37–40]. The authors of these studies proposed that such timing adjustments, which depend on the calls of others, are analogous to primate-like vocal turn-taking. Moreover, call timing was found to reflect social relationships such as pair bonding [40], suggesting that ecological functions of call timing may be more similar across taxa than previously assumed. However, these studies have mainly focused on inter-individual call intervals, and research on intra-individual repeated transitions remains limited (see [41]).

R1-4

Given that I had difficulty interpreting Table 1/Figure 3 initially, the inclusion of the antilogarithmic values is extremely helpful. Please include the x-axis legend on Figure 3, specifying that you are plotting the antilogarithmic values. Additionally, I wonder if reorganizing Table 1 might be helpful by having the columns reflect the short and long components with the mean ± variance in the cells. Then you could more easily present the log transformed and anti-log values for each component.

We appreciate the reviewer’s insightful comments. We have changed the legend of Figure 3 as follows.

Fig 3. Histogram and estimated probability density function curve (curve line) fitted by the two modal GMM for the exchange transition (A) and for repeated transitions (B).

The dashed vertical lines represent the positions of the estimated mean parameter values in the short components of the GMMs. Note that although the data are log-transformed, the x-axis shows antilogarithmic values in seconds.

Aslo, we have re-arranged the Table1 as per the reviewer’s suggestion. In doing so, we realized that variances should not be presented using antilogarithmic values. Therefore, we have removed the antilogarithmic variance values (previously shown in parentheses) from the table.

(see attached file)

Transition

Data size (N)

Components

Mean

Variance

Mixture ratio

Exchange

1710

short

-0.728

(0.187)

0.023

0.322

long

0.321

(2.094)

0.502

0.678

Repeated

3350

short

-0.669

(0.214)

0.035

0.167

long

0.578

(3.784)

0.296

0.833

Overall, modifications to the manuscript have helped showcase how calling behavior in java sparrows differs from those of other birds/mammals to better explain the utility of contagious calling and the value of comparative studies to evaluate how and why different groups of animals generate and modify vocal turn-taking.

R1-5

Minor suggestions/observations:

1) Line 53, clarify ‘it’

We have replaced ‘it’ by ‘the caller’.

R1-6

2) Line 111-112, embed/modify the ‘birds fly much faster than mammals’ – there aren’t many mammalian species that fly, so referring to their movement as ‘travel’ may be more appropriate.

We have made the corrections as per the reviewer’s suggestion.

R1-7

3) Consider combining the sentences on lines 125-127 for readability.

We have changed the expression as per the reviewer’s suggestion.

L. 132-133

There are also short monosyllabic calls, presumed to function as contact calls [44], and we focus on the exchange of these calls.

R1-8

4) Line 178 – replace ‘enhance’ with ‘promote’ or something similar

We appreciate the reviewer’s suggestion and replaced ‘enhance’ by ‘encourage’ and modified the sentence as follows.

L. 182-184

The purpose of the PRE phase was to habituate the participants to the experimental space and encourage spontaneous vocalizations throughout the experimental session while reducing their stress.

R1-9

5) Line 179 – replace ‘incubated’ – “The duration of the F2F phase was 15 minutes.”

We sincerely appreciate the reviewer’s kindness in pointing out the typographical errors. Following the suggestions, I have corrected them accordingly.

R1-10

6) Line 181-2: replace with “During the POST phase, facing individuals were removed from the soundproof room…”

We again sincerely appreciate the reviewer’s kindness. We have corrected the sentence by the suggested phrases in the revised manuscript.

L. 186-187

During the POST phase, facing individuals were removed from the soundproof room with only the preceding individuals remaining, as was the case during the PRE phase.

R1-11

7) Line 390 – use of the phrase, “affected by the vocal behavior of other individuals” doesn’t seem appropriate given that if you’d found differences between the transitions you would have been able to write this exact phrase. Consider clarifying.

We appreciate the reviewer’s comment. To improve clarity, we have removed the sentence and revised the preceding text accordingly.

L. 410-415

In other words, Java sparrows vocally communicated with each other without predicting or anticipating a response from the facing individual, but calling contagiously in response to others’ vocalizations. This is quite different from human conversation, in which turn-taking occurs at an appropriate time by predicting the vocalizations of the person to whom he or she is speaking [1].

R1-12

8) Line 402 – modify to “To more appropriately test our hypothesis we will need to investigate vocal exchange …”

We modified this to rephrase it, along with the comment from Reviewer 2 below.

L. 427-430

To test our hypothesis that contagious calling facilitates the coordination of flock movement, vocal exchange and behavioral synchronization among multiple individuals should be investigated in more natural, flock-like settings.

Reviewer #2:

The authors have sufficiently addressed my previous comments and the revised manuscript is considerably improved.

I would just like to point out the following minor phrasing issues, which can be easily edited prior to publication:

R2-1

Lines 368-369: “Instead, they called in synchrony…” can be too easily misinterpreted as “called simultaneously”, which they apparently do not do in this paradigm, given the low instances of overlap. This new sentence can be removed.

We appreciate the reviewer’s suggestion and have removed the sentence accordingly. However, since we would like to retain the concept of “contagious,” we have revised the preceding sentence as follows.

L. 392-394

Rather, it is likely that Java sparrows responded contagiously to the calls of others, although they did not appear to anticipate a ‘response’ call.

R2-2

Lines 402-404 “To test our hypothesis, we had to investigate the vocal exchanges and behavioral synchronization among more than three individuals in more natural, flock-like settings.”

- This was not done here. Perhaps authors meant “we would have to investigate…”.

-Also “our hypothesis” is a bit confusing here, given that the previous sentence refers to “our study”. Text should specify “To test our hypothesis regarding the synchronization of flocks, we would have to investigate…” (or similar).

We completely agree with the reviewer’s comment. To clarify the meaning of "our hypothesis," we have revised the sentence as follows:

L. 427-430

To test our hypothesis that contagious calling facilitates the coordination of flock movement, vocal exchange and behavioral synchronization among multiple individuals should be investigated in more natural, flock-like settings.

R2-3

Lines 414-415: It is not clear to me that the ability to avoid overlapping with a specific partner or in a playback experiment would necessarily mean that call overlaps would not increase in the presence of more callers. This new sentence can be removed.

We appreciate Reviewer 2’s valuable suggestion to remove the sentence. However, since the anticipation of responses and overlap avoidance are important aspects of vocal exchanges, we have chosen to revise and retain the sentence as follows. We hope Reviewer 2 will understand our intention with this clarification.

L. 436-444

Specifically, in vocal exchanges involving more than two individuals, the chance of call overlap would be expected to increase simply due to the greater number of potential responders. If birds only respond reflexively without anticipating the timing of others' calls, such overlap is likely to occur more frequently. However, if individuals are

---

## [Editor Report · Decision Letter 2]

Temporal regularities of vocal exchange in Java sparrows

PONE-D-24-46388R2

Dear Dr. Koda,

We’re pleased to inform you that your manuscript has been judged scientifically suitable for publication and will be formally accepted for publication once it meets all outstanding technical requirements. Thank you for submitting this interesting paper and for your thorough response to the Reviewers' suggestions. I apologize for the various delays in the editorial process. I look forward to seeing the article published so that others may read it.

Kind regards,

David S Vicario, Ph.D.

Academic Editor

PLOS ONE
---

## [Editor Report · Acceptance letter]

PONE-D-24-46388R2

PLOS ONE

Dear Dr. Koda,

I'm pleased to inform you that your manuscript has been deemed suitable for publication in PLOS ONE. Congratulations! Your manuscript is now being handed over to our production team.

Kind regards,

on behalf of

Dr. David S Vicario

Academic Editor

PLOS ONE